

# Drought and rewatering effects on soybean photosynthesis, physiology and yield

Cheng Wang[1], Anni Sun[2,3], Li jie Zhu[2,4], Min Liu[5], Qi Zhang[6], Liwei Wang[2] and Xining Gao[2]

[1] Liaoning Academy of Agricultural Sciences, Institute of Maize, Shenyang, Liaoning, China
[2] Shenyang Agricultural University, College of Agronomy, Shenyang, Liaoning, China
[3] Anshan Meteorological Bureau, Anshan, Liaoning, China
[4] Chaoyang Meteorological Bureau, Chaoyang, Liaoning, China
[5] Tieling Meteorological Bureau, Tieling, Liaoning, China
[6] Liaoning Provincial Meteorological Information Center, Shenyang, Liaoning, China

## ABSTRACT

Drought stress is a common environmental stress factor for soybeans (*Glycine max* L.), significantly impeding the growth and yield. Therefore, studying the photosynthetic and physiological characteristics during two crucial growth and development periods, namely the flowering and grain-filling stages, under drought stress and rewatering conditions is of great significance for clarifying the physiological and photosynthetic regulatory response mechanisms of soybeans to drought stress. In this study, the cultivar 'Liaodou 15' was subjected to mild drought (L, 65% field capacity) and severe drought (H, 50% field capacity) treatments during the flowering and grain-filling stages for 7, 14, and 21 days respectively. At the conclusion of the stress period, rewatering (R) was carried out. Results showed that the stomatal limit value increased and intercellular $CO_2$ concentration decreased with the increase in drought stress intensity, and the decrease in net photosynthetic rate was dominated by stomatal factors at the flowering stage. At the grain-filling stage, the stomatal limit value decreased and intercellular $CO_2$ concentration increased with the increase in drought stress intensity, and the decrease in net photosynthetic rate changed from stomatal factors to non-stomatal factors. Drought stress led to peroxidation damage. In this study, it significantly increased the contents of soluble protein and malondialdehyde (MDA), as well as the activities of peroxidase (POD) and superoxide dismutase (SOD). On the other hand, rewatering had a compensatory effect on various physiological indices of soybean leaves. Under drought stress, the yield indices of soybeans were affected during both the flowering and grain-filling stages. Specifically, the yield during the flowering stage decreased by 15.63%–55.47%, and the yield during the grain-filling stage decreased by 24.17%–59.63%. This indicates that drought has a greater impact on the yield of soybeans during the grain-filling stage. Moreover, as the duration and intensity of drought increase, the reduction in yield becomes more significant, and the yield is the lowest when there is severe drought stress for 21 days. Our study elucidates the complex physiological and photosynthetic responses of soybeans to drought stress and rewatering, which provides valuable insights for improving soybean cultivation strategies under drought environments.

Corresponding authors
Liwei Wang, wlw@syau.edu.cn
Xining Gao, syaugxn@syau.edu.cn

# INTRODUCTION

Soybean (*Glycine max* L.) is the main oil crop in the world. Northeast China has the largest soybean production base and great production potential. However, soybean is susceptible to water deficit, and drought seriously affects this crop (*Liu et al., 2005*). A comprehensive assessment of how droughts with complex patterns during the period from 1,981 to 2016 have continuously exerted negative impacts on crop yields globally reveals that the yield of soybeans is vulnerable under complex drought patterns, and when the degree of drought shifts from moderate to extreme, the yield loss of soybeans significantly intensifies (*Santini et al., 2022*). Under high levels of global warming and more extreme climate events, the frequency and intensity of drought are becoming increasingly serious, severely impacting agricultural production (*IPCC, 2021*). Drought is the factor that has the greatest impact on crop yield among all environmental stresses. Compared with other natural disasters, drought occurs more frequently, lasts longer, and creates a wide range of impacts. The loss caused by drought is almost equal to the total loss caused by all other environmental factors (*Bhat et al., 2020*; *Biji et al., 2008*; *Chandra et al., 2021*).

Drought stress results in water shortage in leaf cells and turgor pressure reduction, thus inhibiting the elongation and growth of cells and affecting the physiological characteristics and growth and development of crops, which is a main obstacle affecting crop yield (*Rekika et al., 1998*; *Saidou, Janssen & Temminghoff, 2003*). Drought stress can severely damage the plant cell membrane system. The stability of the cell membrane is crucial for the orderly conduct of cellular life activities. During drought, cytoplasmic dehydration disrupts membrane permeability, causing cell damage. Drought also induces the generation of reactive oxygen species (ROS), including superoxide, alkoxyl, and hydroxyl radicals, as well as non-radical substances like hydrogen peroxide. These are highly toxic and reactive, disrupting intracellular homeostasis by damaging proteins, carbohydrates, lipids, and DNA (*Farooq et al., 2012*). To cope with drought, plants activate intricate defense mechanisms, including osmotic, hormonal, metabolic, and redox regulations. Osmotic regulation represents a vital adaptive strategy under stress, enabling plants to mitigate damage. By accumulating osmoprotectants such as proline, betaine, and soluble sugars, plants lower the intracellular water potential, thereby countering drought stress through osmotic adjustment (*Noreen, Athar & Ashraf, 2013*; *Kaur & Asthir, 2017*). Plants also respond to drought by modulating hormone levels. For instance, abscisic acid (ABA) promotes stomatal closure and inhibits opening, minimizing water loss. Adequate ethylene, on the other hand, stimulates root growth and development, resulting in a more robust root system that can absorb more water from the soil (*Christmann et al., 2005*; *Takahashi et al., 2018*). Metabolically, plants synthesize and accumulate secondary metabolites like flavonoids, phenols, and lignin. These substances enhance the strength and toughness of the plant cell wall, thereby improving plant stress tolerance (*Winkel-Shirley, 2001*). Redox

regulation is a key strategy for plants to combat drought. To eliminate ROS induced by drought stress, plants employ two distinct detoxification mechanisms: enzymatic reactions and non-enzymatic antioxidants. This effectively reduces or prevents damage to plant cells caused by excessive ROS (*Suzuki et al., 2014*). Numerous studies have demonstrated that under stress conditions, plants significantly increase the production of various protective substances, such as superoxide dismutase (SOD), peroxidase (POD), catalase (CAT), and ascorbate peroxidase (APX) (*Osakabe et al., 2014*; *Zia et al., 2021*).

Drought stress not only affects the physiological indexes of plants but also greatly impacts photosynthesis (*Song, Zhou & He, 2021*; *Wang et al., 2021*). Severe drought stress leads to metabolic disorders in plants, eventually leading to their death (*Jaleel et al., 2008*). Drought stress limits plant growth by reducing photosynthetic rates (*Kebbas, Lutts & Aid, 2015*; *Zhang et al., 2018*). The main factors for the decrease in photosynthesis may be stomatal and non-stomatal factors caused by decreased carbon dioxide ($CO_2$) and photosynthetic activity in mesophyll tissue, respectively (*Song, Zhou & He, 2021*). There is a significant correlation between stomatal conductance and photosynthetic response under drought stress, indicating that stomatal conductance plays a major role in decreasing the photosynthetic rate in leaves (*Varone et al., 2012*; *Ghotbi-Ravandi et al., 2014*).

The changes in physiological and photosynthetic indices caused by drought will severely affect the growth and development of plants. All stages from seed germination, vegetative growth to reproductive growth are hindered, resulting in stunted plants, reduced leaf area, abnormal flower bud differentiation, poor pollination, and increased susceptibility to pests and diseases. Eventually, the number of fruits or grains of the plants decreases, the quality deteriorates, and the yield drops significantly (*Lipiec et al., 2013*; *Oguz et al., 2022*).

Rewatering after drought stress can further reduce the damage caused by drought stress to a certain extent. The mechanisms involve osmotic adjustment, photosynthetic compensation and reactive oxygen species scavenging. Under drought, plants lower osmotic potential. After rewatering, its recovery lags behind that of leaf water potential, allowing crops to maintain high osmotic adjustment capacity for a long time to compensate for drought losses (*He et al., 2024*). Regarding photosynthetic efficiency, during drought, stomatal aperture narrows, transpiration rate drops and photosynthetic water use efficiency rises significantly. After rewatering, stomatal aperture stays low, photoinhibition eases, photosynthetic rate recovers and photosynthetic water use efficiency further improves (*Escalona, Flexas & Medrano, 2000*). Drought stress generates a lot of reactive oxygen species in crops, increasing the activities of SOD, POD, *etc.* After rewatering, these enzyme activities must stay high for a while to enhance reactive oxygen species scavenging (*Mu et al., 2021*). Different crops and degrees of drought stress after rewatering have considerable differences in compensation effects (*Mu et al., 2021*).

Previous studies have investigated the effects of drought stress on physiological indexes, such as osmotic regulatory substances and antioxidant enzymes, and photosynthetic characteristics of soybean (*Wang et al., 2018*; *Buezo et al., 2019*). However, due to the varying effects of drought stress intensity and duration and degree of recovery after rewatering on physiological and photosynthetic characteristics and yield of soybean, there were significant differences in the results caused by these different research studies.

Most studies adopted pot experiments that achieved drought stress in a relatively short time through a rapid decline in soil moisture. However, during the actual production process, drought is a slow accumulating process; therefore, the response mechanisms of physiological and photosynthetic characteristics to drought stress are different from those under pot conditions. Drought has different impacts on soybeans at various growth stages. If soybeans encounter drought during the flowering stage, it will disrupt the hormonal balance, affect the differentiation and development of flower buds, reduce the viability of pollen, and influence the formation of fruits. When drought occurs during the grain-filling stage, it will hinder the transportation of photosynthetic products, affect the development of grains, and lead to yield losses (*Xie et al., 1994*; *Wei et al., 2018*). It's been widely reported that reproductive stages are more drought-susceptible to drought than vegetative stages, especially the flowering and grain-filling stage. Therefore, in this study, two key growth stages that have a significant impact on soybean yield, namely the flowering stage and the grain-filling stage, were selected as the research subjects. The water control experiment of the sliding canopy was used in this study. It aimed to study the impacts of the time at which drought occurs during the flowering and grain-filling stages, as well as drought stress with different durations and intensities and rewatering on photosynthetic characteristics, physiological characteristics, and yield of soybean under field conditions, and to analyze the physiological response mechanism. The results of this study will lay a foundation for a more in-depth analysis of the physiological and photosynthetic regulatory mechanisms of soybean under drought and rewatering conditions, and provide a reference and theoretical basis for improving soybean cultivation strategies in arid environments.

## MATERIALS & METHODS

### Experimental materials and design

The field experiment was establshed in 2020 at the Scientific Observation and Experiment Station of Crop Cultivation of the Ministry of Agriculture and Rural Affairs in the northeast region, Liaoning province, China (41.73°N, 123.53°E). The test station was a large-scale water control test site with flat terrain and brown soil. Its nutrient contents were as follows: 15.78 g kg$^{-1}$ of organic matter, l.18 g kg$^{-1}$ of total nitrogen, 0.55 g kg$^{-1}$ of total phosphorus, 21.68 g kg$^{-1}$ of total potassium, 8.52 mg kg$^{-1}$ of available phosphorus, 98.20 mg kg$^{-1}$ of available potassium, 67.30 mg kg$^{-1}$ of alkaline-hydrolyzed nitrogen, and the pH was 6.42. The soil bulk density was 1.25 g cm$^{-3}$, and the field water capacity was 30%.

The experimental soybean variety was Liaodou 15, widely cultivated in the Liaoning province of China, and bred by the Liaoning Academy of Agricultural Sciences. The fertilizer applied in the planting field was the local conventional fertilization level, and 14-16-15 compound fertilizer (45%) was adopted. Nitrogen fertilizer was applied at 52.5 kg hm$^{-2}$ (N), phosphate fertilizer at 60 kg hm$^{-1}$ (P$_2$O$_5$), and potassium fertilizer at 56.25 kg hm$^{-1}$ (K$_2$O) during the whole growth period of soybean. The fertilizer was applied once as base fertilizer before sowing in spring. The seeds were sown on May 4, the flowering period began on July 9, the drum period began on August 12, and the soybeans were harvested on September 27.

The sliding canopy and artificial water supply were used to control soil moisture during the whole growth period. Drip irrigation was used to refill water, and monitor the soil water content by using a TDR (Time Domain Reflectometry) moisture detector, and conduct water control in combination with a flowmeter. Two key growth and development periods that have a significant impact on the soybean yield, namely the flowering stage and the grain-filling stage, were selected as the research objects. The water control experiment was carried out when the soybean growth stage entered the flowering (labeled F) and grain-filling (labeled S) stages. Under the conditions of mild (65% field capacity, labeled L) and severe drought stress (50% field capacity, labeled H) drought stresses, the drought lasted for 7, 14, and 21 days (labeled 07, 14, and 21, respectively). At the end of the stress duration, rewatering was performed (labeled R), and rewatered to the control level (80% field capacity, labeled CK). There were 12 treatments in the experiment, with triplicates per treatment and triplicates for the control. A total of 39 plots were set up. The plot area was 12 m$^2$, the planting ridge length was 2 m, the inter-row spacing was 0.6 m, and the intra-row spacing was 0.11 m. The planting method was acupoint sowing, and the seedling density was 180 plants per plot. Sampling of plants was carried out in a random manner.

## Measuring items and methods
### Measurement of yield index
During the soybean harvest period, three ridges were randomly selected from each plot, and 10 consecutive plants were selected from the middle position of each ridge as the objects for measuring yield indicators. After drying the seeds, yield attributes, including the number of grains per plant, number of blighted grains per plant, number of pods per plant, and crop yield, were measured at the crop maturity in triplicates.

### Measurement of physiological indexes
Measurements were performed at 9:00 am the day after the duration of drought stress treatments were completed and one week after rewatering. Three representative plants with uniform growth were selected from each plot for measurement, and the penultimate leaf was measured for each plant. After sampling, the leaves were frozen in liquid nitrogen and stored at −80 °C for physiological measurements. SOD activity was determined using the nitroblue tetrazolium (NBT) method (*Alici & Arabaci, 2016*), while POD activity was measured using the guaiacol method (*Cazenave et al., 2006*). In addition, MDA (malondialdehyde) and soluble protein contents were measured using the thiobarbituric acid (TBA) (*Schmedes & Hølmer, 1989*) and the Coomassie bright blue (*Read & Northcote, 1981*) methods, respectively.

### Measurement of photosynthetic parameters
Measurements were performed between 9:00 am and 12:00 noon under atmospheric $CO_2$ the day after drought stress treatment and one week after rewatering. Three representative plants with uniform growth were selected from each plot for measurement, and the penultimate leaf was measured for each plant. Photosynthesis-light response curves of net photosynthesis (Pn, $\mu$mol m$^{-2}$ s$^{-1}$), stomatal conductance (Gs, $\mu$mol mol$^{-1}$), and internal $CO_2$ concentration (Ci, $\mu$mol mol$^{-1}$), were determined using a portable gas exchange

measuring system (Li6400XT, Li-Cor, USA). The $CO_2$ concentration was set to 400 μmol mol$^{-1}$, and photosynthetically active radiation (PAR) was set to 2,000, 1,800, 1,500, 1,200, 1,000, 800, 600, 400, 200, 100, 70, 40, 20, 10, and 0 μmol m$^{-2}$ s$^{-1}$. Photosynthesis measurements were made at a block temperature of 30.0 °C. Other parameters were calculated according to the following equations: stomatal limitation: LS = 1−Ci/Ca. The apparent quantum efficiency (AQE) and maximum photosynthetic rate (Pmax, μmol m$^{-2}$ s$^{-1}$) in the light response curve were obtained by fitting the logistic model: Pn = AQE × PAR × Pmax/(AQE × PAR + Pmax) − Rday (Rday is the respiratory rate, μmol m$^{-2}$ s$^{-1}$).

### Data processing

One-way analysis of variance (ANOVA) with the least significant difference (LSD) test at $P < 0.05$ was used to analyze the differences of leaf physiological and photosynthetic characteristics and yield of soybean under drought treatments. SPSS25 was used to fit the logistic model and perform multiple comparative analyses.

## RESULTS

### Effects of drought stress and rewatering on yield

Drought stress significantly reduced the number of grains per plant, number of pods per plant and yield, and significantly increased the number of bligthed grain per plant. At the flowering stage, compared with CK, number of grains per plant declined by 12.23%, 25.70%, 23.51%, 33.53%, 32.60%, and 55.47%, and the yield declined by 15.63%, 27.25%, 24.69%, 34.07%, 36.53%, and 55.47% under L-07, H-07, L-14, H-14, L-21, and H-21, respectively. At the grain-filling stage, compared with CK, number of grains per plant declined by 22.25%, 29.46%, 33.86%, 37.93%, 42.32%, and 49.84%, and the yield declined by 24.17%, 32.36%, 37.07%, 43.39%, 43.63%, and 559.63% under L-07, H-07, L-14, H-14, L-21, and H-21, respectively. With the increase in drought duration and intensity, the yield indicators decreased more, and the yield under severe drought stress for 21 days was the lowest (Table 1).

The results indicated that, except for the number of pods per plant, the impact of drought on other yield indicators was greater during the grain-filling stage than during the flowering stage. As the duration and intensity of drought increased, the yield decreased more significantly.

### Effects of drought stress and rewatering on SOD activity

At the flowering stage, the SOD activity under mild drought stress increased, while it decreased under severe drought stress compared with CK. The SOD activity under FL-07, FL-14 and FL-21 was significantly increased by 36.69%, 29.17% and 18.63%, respectively, compared with CK. While under FH-21, it significantly reduced by 7.95%, compared with CK. Under drought stress, the SOD activity of soybean at the grain-filling stage was similar to that at the flowering stage. Compared with CK, the SOD activity significantly increased by 12.31%, 8.82%, 16.41% and 25.90%, respectively under SL-07, SH-07, SL-14 and SL-21. While under SH-21, it reduced by 1.80% compared with CK. The SOD activity decreased

Wang et al. (2025), *PeerJ*, DOI 10.7717/peerj.19658

**Table 1  Multiple comparison of soybean yield under the drought stress treatments.**

| Growth period | Yield index | CK | Treatment | | | | | |
| --- | --- | --- | --- | --- | --- | --- | --- | --- |
| | | | L-07 | H-07 | L-14 | H-14 | L-21 | H-21 |
| Flowering stage | Number of grains per plant | 106.33 ± 2.52a | 93.33 ± 1.53ab | 79.00 ± 1.00b | 81.33 ± 0.58b | 70.67 ± 1.53bc | 71.67 ± 2.08bc | 55.33 ± 2.08c |
| | Number of bligthed grain per plant | 9.67 ± 0.58b | 11.00 ± 1.00b | 14.67 ± 1.15ab | 12.00 ± 1.00b | 17.67 ± 0.58a | 13.33 ± 0.58b | 19.33 ± 1.15a |
| | Number of pods per plant | 57.67 ± 2.52a | 50.33 ± 2.31ab | 51.67 ± 3.06ab | 46.00 ± 1.00b | 43.33 ± 1.53b | 46.67 ± 1.15b | 33.33 ± 1.53c |
| | Yield (kg hm$^{-2}$) | 4,525.20 ± 68.12a | 3,817.80 ± 82.18b | 3,292.05 ± 64.02bc | 3,407.85 ± 152.98bc | 2,983.50 ± 106.86c | 2,872.01 ± 79.10c | 2,015.22 ± 67.38d |
| Grain-filling stage | Number of grains per plant | 106.33 ± 2.52a | 82.67 ± 2.52b | 75.00 ± 1.00b | 70.33 ± 0.58bc | 66.00 ± 0.00c | 61.33 ± 1.53cd | 53.33 ± 1.15d |
| | Number of bligthed grain per plant | 9.67 ± 0.58b | 11.00 ± 1.00b | 14.67 ± 1.15b | 12.00 ± 1.00b | 18.67 ± 1.53a | 18.67 ± 1.53a | 21.00 ± 1.00a |
| | Number of pods per plant | 57.67 ± 2.52a | 53.33 ± 2.31a | 52.67 ± 1.15a | 40.33 ± 2.31b | 41.33 ± 1.15b | 34.67 ± 0.58b | 36.00 ± 1.00b |
| | Yield (kg hm$^{-2}$) | 4,525.20 ± 68.12a | 3,431.40 ± 69.14b | 3,061.05 ± 83.76bc | 2,847.60 ± 39.31c | 2,561.85 ± 91.10c | 2,551.05 ± 69.30c | 1,827.00 ± 47.23d |

Notes.

L means light drought; H means severe drought; 07, 14 and 21 means the treatments lasted for 7, 14 and 21 days; R means rewatering. The data is mean with standard deviation. Lower case means the difference among the drought treatments is significant ($P < 0.05$).

**Table 2  Comparisons of the physiological index under drought stress and rewatering.**

| | Growth period | Treatment | Treatment of drought stress and rewatering | | | | | |
|---|---|---|---|---|---|---|---|---|
| | | | 07 | 07-R | 14 | 14-R | 21 | 21-R |
| SOD activity (U g⁻¹ FW min⁻¹) | Flowering stage | CK | 530.58 ± 24.77bA | 505.83 ± 21.12a | 505.83 ± 21.12bAB | 482.52 ± 6.88a | 482.52 ± 6.88bB | 473.3 ± 21.15a |
| | | FL | 725.24 ± 21.84aA | 484.95 ± 12.44a** | 653.40 ± 11.68aB | 467.96 ± 11.12a** | 570.39 ± 5.11aC | 447.57 ± 15.71ab** |
| | | FH | 524.27 ± 16.79bA | 406.8 ± 11.68b** | 490.24 ± 6.50bB | 423.79 ± 18.25b** | 444.17 ± 10.19bC | 437.38 ± 14.14b** |
| | Grain-filling stage | CK | 473.3 ± 21.15bA | 440.78 ± 14.59a | 440.78 ± 14.59bAB | 404.85 ± 13.35a | 404.85 ± 13.35bB | 391.75 ± 14.04a |
| | | SL | 531.55 ± 11.65aA | 426.7 ± 10.19a** | 513.11 ± 14.59aA | 390.78 ± 8.90ab** | 509.71 ± 6.35aA | 386.89 ± 15.30a** |
| | | SH | 431.55 ± 15.44cA | 373.3 ± 13.99b** | 423.3 ± 12.72bA | 375.73 ± 7.57b** | 397.57 ± 32.20bA | 380.1 ± 10.50a |
| POD activity (U g⁻¹ FW min⁻¹) | Flowering stage | CK | 3.03 ± 0.04aa | 3.35 ± 0.05a** | 3.35 ± 0.05aB | 3.67 ± 0.07a** | 3.67 ± 0.07aC | 4.07 ± 0.08a** |
| | | FL | 3.62 ± 0.06bA | 3.92 ± 0.07b** | 3.83 ± 0.04bB | 4.22 ± 0.11b** | 4.10 ± 0.09bC | 4.57 ± 0.04b** |
| | | FH | 3.83 ± 0.03cA | 4.49 ± 0.15c** | 4.03 ± 0.07cB | 4.64 ± 0.06c** | 4.31 ± 0.04cC | 4.91 ± 0.07c** |
| | Grain-filling stage | CK | 4.07 ± 0.08bA | 3.71 ± 0.05b** | 3.71 ± 0.05aB | 3.41 ± 0.05a** | 3.41 ± 0.05aC | 3.41 ± 0.03a** |
| | | SL | 4.42 ± 0.03aA | 3.98 ± 0.04a** | 3.43 ± 0.04bB | 3.42 ± 0.04a | 3.16 ± 0.08bC | 3.41 ± 0.03a* |
| | | SH | 3.95 ± 0.06cA | 3.63 ± 0.04b | 3.22 ± 0.03cB | 3.38 ± 0.05a** | 2.92 ± 0.06cC | 3.40 ± 0.03a** |
| MDA content (nmol g⁻¹) | Flowering stage | CK | 28.24 ± 1.54aA | 29.80 ± 2.15a | 29.80 ± 2.15aAB | 32.60 ± 2.63a | 32.60 ± 2.63aB | 35.66 ± 2.28a |
| | | FL | 33.92 ± 0.70bA | 31.26 ± 1.26ab* | 43.26 ± 1.93bAB | 35.93 ± 2.30ab* | 49.03 ± 1.87bC | 38.22 ± 2.14ab** |
| | | FH | 39.53 ± 2.03cA | 34.290 ± 1.24b* | 53.19 ± 1.69cB | 39.12 ± 4.32b** | 57.90 ± 2.870cC | 41.330 ± 1.38b** |
| | Grain-filling stage | CK | 35.66 ± 2.28aA | 38.43 ± 2.76a | 38.43 ± 2.76aAB | 41.863 ± 2.37a | 41.86 ± 2.37aC | 44.15 ± 1.87a |
| | | SL | 52.70 ± 1.72bA | 41.50 ± 2.37ab** | 57.57 ± 2.37bB | 45.47 ± 1.57a** | 63.18 ± 1.71bC | 51.82 ± 2.89b** |
| | | SH | 62.45 ± 2.37cA | 43.853 ± 2.00b** | 68.504 ± 2.29cB | 58.54 ± 1.92b** | 74.13 ± 2.14cC | 58.94 ± 0.85c** |
| Soluble protein content (μ mol g⁻¹) | Flowering stage | CK | 16.07 ± 0.58aA | 16.49 ± 0.63a | 16.49 ± 0.63aA | 16.78 ± 0.52a | 16.78 ± 0.52aA | 16.81 ± 0.25a |
| | | FL | 17.10 ± 0.28bA | 17.05 ± 0.38a | 17.39 ± 0.23bAB | 17.05 ± 0.29a | 17.82 ± 0.18bB | 17.18 ± 0.19ab* |
| | | FH | 17.77 ± 0.25bA | 17.18 ± 0.22a* | 17.98 ± 0.27bAB | 17.30 ± 0.29a* | 18.43 ± 0.27bB | 17.85 ± 0.18b* |
| | Grain-filling stage | CK | 16.81 ± 0.25aA | 17.09 ± 0.24a | 17.09 ± 0.24aAB | 17.32 ± 0.24a | 17.32 ± 0.24aB | 17.49 ± 0.30a |
| | | SL | 18.13 ± 0.22bA | 17.42 ± 0.27a* | 18.38 ± 0.37bA | 17.57 ± 0.21ab* | 18.65 ± 0.27bA | 18.46 ± 0.28b |
| | | SH | 18.82 ± 0.36cA | 17.63 ± 0.39a* | 19.03 ± 0.30cA | 18.20 ± 0.21b* | 19.26 ± 0.35cA | 18.69 ± 0.35bc |

**Notes.**

L means light drought; H means severe drought; 07, 14 and 21 means the treatments lasted for 7, 14 and 21 days; R means rewatering. The data is mean with standard deviation. Lower case means the difference among the drought treatments is significant ($P < 0.05$), and capital letter means the difference among drought days is significant ($P < 0.05$).

*Denotes a significant difference between the drought treatment and rewatering groups ($P < 0.05$).

**Denotes an extremely significant difference between the two groups ($P < 0.01$).

with the duration of drought stress. The SOD activity under FL-14 and FL-21 decreased by 9.91% and 21.35%, respectively, compared with the FL-07 treatment, while that of the FH-14 and FH-21 treatments decreased by 6.49% and 15.28%, respectively, compared with the FH-07 treatment. At the same intensity of drought stress at the flowering stage, the SOD activity was significantly different at different duration. The trend of SOD activity decreased with the duration of drought stress at the grain-filling stage and was similar to that at the flowering stage (Table 2).

The SOD activity decreased after rewatering at the flowering stage. The SOD activity under FL-07-R and FH-07-R decreased by 4.13% and 19.58% compared with CK, respectively, showing overcompensation. The SOD activity under FL-14-R, FH-14-R, FL-21-R and FH-21-R decreased by 3.02%, 12.17%, 5.44% and 7.59% compared with CK, respectively. At the flowering stage, there were significant differences in SOD activity before and after rewatering at 7 and 14 days under drought stress. The difference at 21 days

under mild drought stress reached an extremely significant level. At the grain-filling stage, The SOD activity under SL-07-R and SH-07-R decreased by 3.19% and 15.31% compared with CK, respectively. Similarly, the SOD activity after SL-14-R, SH-14-R, SL-21-R, and SH-21-R treatments decreased compared with CK (Table 2). Therefore, the SOD activity of the above drought-stress rewatering treatments showed a compensation effect.

The results indicated that, during the flowering and grain-filling stages, the change trends of the SOD activity in soybeans under drought stress were similar, which varied with the stress intensity and duration. Mild drought often led to an increase in its activity, while severe drought caused a decrease. After rewatering, a compensatory effect was observed, and there were significant differences in the SOD activity at each stage under different treatments.

## Effects of drought stress and rewatering on POD activity

The POD activity of soybean under drought stress at the flowering stage showed an increasing trend. Compared with CK, the POD activity under FL-07, FH-07, FL-14, FH-14, FL-21, and FH-21 increased significantly by 19.35%, 26.31%, 14.33%, 20.36%, 11.71%, and 17.52%, respectively. The POD activity increased more under severe drought stress at the flowering stage. The POD activity increased by 8.67% under SL-07 but decreased under other treatments. Compared with CK, all treatments reached the significance level, and the decreasing range increased with the stress intensity. With the increase in drought duration, the POD activity increased at the flowering stage and decreased at the grain-filling stage. The POD activity under FL-14 and FL-21 significantly increased by 5.91% and 13.44% compared with that under FL-7. In FH-14 and FH-21 treatments, it significantly increased by 5.36% and 12.75% compared with FH-7, respectively. The POD activity under SL-14 and SL-21 significantly decreased by 22.49% and 28.50% compared with SL-7, and under SH-14 and SH-21, it significantly decreased by 18.34% and 25.94% compared with SH-7, respectively (Table 2).

After rewatering under drought stress, the POD activity at the flowering stage still showed an increasing trend. Compared with CK, the FL-07-R, FH-07-R, FL-14-R, FH-14-R, FL-21-R and FH-21-R treatments significantly increased the POD activity by 17.11%, 33.92%, 14.87%, 26.51%, 12.21%, and 20.49%, respectively. The POD activity showed a partial compensation effect after rewatering under mild drought stress. Compared with CK, the POD activity under SL-07-R was significantly increased by 7.08%, while that after other treatments were not significantly different at the grain-filing stage, showing equal compensation. After drought stress for 7 and 21 days, the POD activity of all treatments was significantly different from that before and after rewatering. However, after 14 days of drought stress, only severe drought stress showed a significant difference in POD activity (Table 2).

The results indicated that, during the flowering stage, drought increased POD activity, especially under severe drought. In the grain-filling stage, POD activity decreased in most treatments. After rewatering, it rose in the flowering stage but declined in the grain-filling stage. After rewatering, POD activity in the flowering stage kept rising, with a compensatory effect under mild drought.

## Effects of drought stress and rewatering on MDA content

The MDA content of soybean increased significantly under drought stress. At the flowering stage, the MDA content under FL-07, FH-07, FL-14, FH-14, FL-21, and FH-21 was significantly increased by 20.00%, 40.00%, 45.13%, 78.52%, 50.46%, and 77.61%, respectively, compared with CK. At the grain-filling stage, the MDA content under SL-07, SH-07, SL-14, SH-14, SL-21, and SH-21 was significantly increased by 47.83%, 75.18%, 49.67%, 78.15%, 51.02%, and 77.18%, respectively, compared with CK. Under the same drought stress intensity, the MDA content significantly increased with the drought stress duration. Compared with the FL-07, the MDA content under FL-14 and FL-21 increased by 27.54% and 44.55%, respectively, while that of FH-14 and FH-21 increased by 34.56% and 46.47% compared to FH-07, respectively. The MDA content significantly increased under 14 and 21 days of drought stress compared with 7 days at the grain-filling stage (Table 2).

After rewatering under drought stress, the difference in MDA content between treatments and CK gradually decreased, showing a compensatory effect. At the flowering stage, compared with CK, the MDA content under FL-07-R, FH-07-R, FL-14-R, FH-R-14, FL-21-R, and FH-21-R increased by 4.87%, 15.10%, 10.28%, 19.94%, 7.15%, and 15.99%, respectively. After rewatering under mild drought stress, the MDA content showed an equal compensation effect, while under severe drought stress, the MDA content showed a partial compensation effect. At the grain-filling stage, the MDA content under SL-07-R and SH-07-R increased by 7.93% and 14.04%, respectively, compared with CK, with equal compensation during mild drought stress and partial compensation during severe drought stress. Compared with CK, the MDA content under SL-14-R, SL-21-R, and SH-21-R increased by 8.60%, 17.32%, and 33.30%, respectively, showing a partial compensation effect. However, the MDA content under SH-14-R significantly increased by 39.90% compared with CK, and the compensation effect was low. At the flowering and grain-filling stages, the MDA content before and after rewatering under drought stress showed a significant difference (Table 2).

The results showed that under drought stress, the MDA content of soybeans during both the flowering and grain-filling stages increased significantly, and the greater the stress intensity and the longer the duration, the more substantial the increase. After rewatering the differences in MDA content between each treatment and the control gradually decreased, exhibiting a compensatory effect.

## Effects of drought stress and rewatering on soluble protein content

The content of soluble protein in soybean leaves increased under drought stress. Compared with CK, the FL-07, FH-07, FL-14, FH-14, FL-21, and FH-21 treatments increased soluble protein content by 6.41%, 10.58%, 5.46%, 9.36%, 6.20%, and 9.83%, respectively, and the SL-07, SH-07, SL-14, SH-14, SL-21, and SH-21 treatments increased it by 7.26%, 11.96%, 7.14%, 11.35%, 7.68%, and 11.20%, respectively. When the duration of drought stress was the same, the difference in soluble protein content between mild and severe stresses reached a significant level. Under mild or severe drought stresses, the soluble protein content at the flowering and grain-filling stages was positively correlated with the stress days. The soluble

protein content decreased after rewatering under drought stress compared with that before rewatering. The soluble protein content under FL-21-R, FH-07-R, FH-14-R, and FH-21-R decreased significantly, showing a compensative effect (Table 2). The results indicated that drought stress induced an increase in the soluble protein content of soybean leaves, and rewatering after drought could exhibit a compensatory effect at different levels.

## Effects of drought stress and rewatering on stomatal conductance

$C_i$ and $L_s$ are important criteria to determine the reduction of leaf $P_n$ by stomatal and non-stomatal factors. The decrease in $C_i$ and the increase in $L_s$ indicated that the stomatal factor was the main reason for the decrease in $P_n$; on the contrary, the decrease in $P_n$ was mainly due to non-stomatal factors (*Jones, 1985*). There was no significant difference in $L_s$ and $C_i$ values among CK, FL-07, and FH-07 treatments at the same PAR level, while under 14 and 21 days of drought stress, with the increase in drought stress intensity, the $L_s$ value increased and $C_i$ value decreased, and a similar trend was found after rewatering of different drought stress-treated groups (Fig. 1). This indicated that the effect of drought stress and rewatering on $P_n$ is mainly due to stomatal conductance. Under the same PAR level, compared with CK, the $L_s$ value increased under SL-07, SH-07, SL-14, and SH-14, and the $C_i$ value decreased, while with the increase in drought stress intensity, the $L_s$ value decreased and $C_i$ value increased under SL-21 and SH-21, and a similar trend was found after rewatering of different drought stress-treated groups (Fig. 1). The results indicated that the effects of drought stress and rewatering on $P_n$ during both the flowering and grain-filling stages were mainly caused by stomatal conductance. However, during the grain-filling stage, as the duration of stress increased, the influencing factors of $P_n$ started to shift, with non-stomatal limitations becoming dominant.

## Effects of drought stress and rewatering on Pn

Under FL-14, FH-14, FL-21, and FH-21, $P_n$ decreased at the same PAR level. Compared with CK, AQE under FL-14 and FH-14 decreased by 12.90% and 18.28%, and Pmax decreased by 19.56% and 30.00%, respectively. In addition, AQE under FL-21 and FH-21 decreased by 28.72% and 29.36%, and Pmax decreased by 42.66% and 70.86%, respectively. The AQE and Pmax were recovered after rewatering under 14 and 21 days of drought stress but did not reach CK level (Fig. 2). At the same PAR level, the $P_n$ decreased under drought stress at the grain-filling stage. Compared with CK, AQE decreased by 19.11% and 21.15%, and Pmax decreased by 44.38% and 48.21% under SL-07 and SH-07; AQE decreased by 24.20% and 24.29%, and Pmax decreased by 56.72% and 58.11% under SL-14 and SH-14. Furthermore, AQE decreased by 30.24% and 31.09%, and Pmax decreased by 73.72% and 74.03% under SL-21 and SH-21. AQE and Pmax were recovered after rewatering but did not reach CK level (Fig. 2). Research findings indicated that short-term drought during the flowering stage didn't notably affect photosynthetic efficiency related indexes, but long-term drought caused significant declines and they didn't reach the CK level after rewatering. In the grain-filling stage, drought stress significantly reduced these indexes and there was no recovery to the CK level after rewatering.

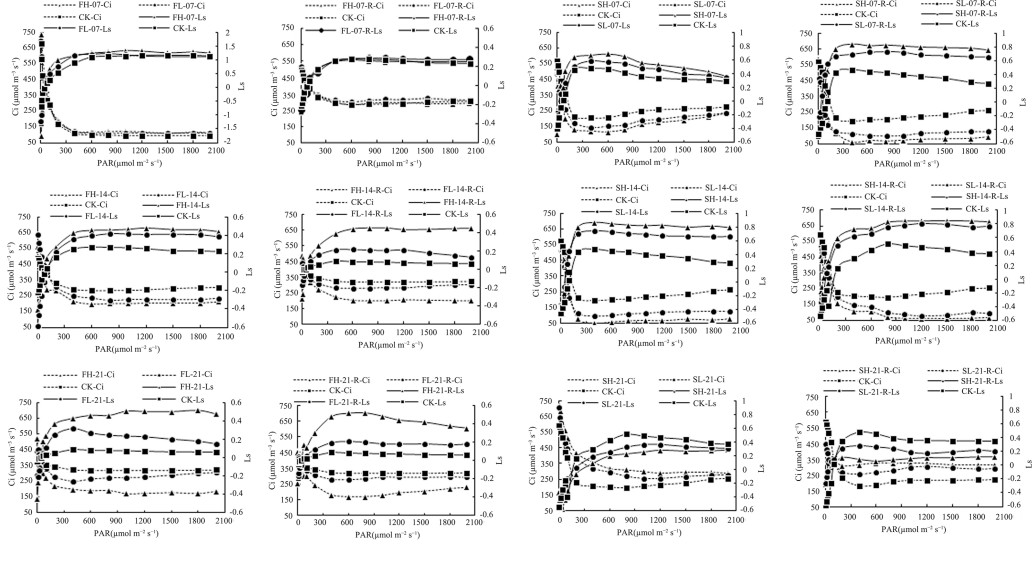

**Figure 1** Ci and Ls of soybean under drought stress and rewatering.

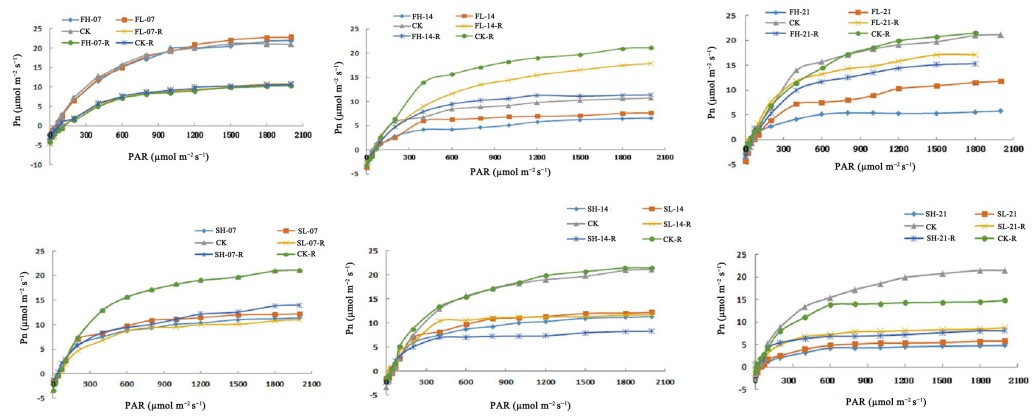

**Figure 2** Pn of soybean under drought stress and rehydration.

## DISCUSSION

Against the backdrop of global climate change, the frequency and intensity of drought events are on the rise, posing a severe challenge to the growth and development of plants in terrestrial ecosystems. As a major environmental stress factor, drought stress has a direct and profound impact on the physiological metabolism and photosynthesis of plants. Understanding how drought stress affects plant photosynthesis not only helps us gain a deeper insight into the survival strategies of plants in adverse conditions but also holds broad and significant implications for agricultural production, the stability of ecosystems, and the global carbon cycle. The Pn is an important index of photosynthesis, and the decrease in photosynthetic rate is caused by stomatal and non-stomatal factors

(*Song, Zhou & He, 2021*; *Wang et al., 2021*). Previous studies showed that when crops were subjected to mild drought stress, stomatal restriction occurred in leaves, leading to the partial closure of stomata in leaves, insufficient supply of $CO_2$ substrate, and a decrease in the net photosynthetic rate (*Jones, 1985*). Under severe stress, non-stomatal restriction occurred in the leaves. Drought damaged the chloroplast function and reduced the net photosynthetic rate. The Pn could not be effectively restored even after rewatering (*Kanechi et al., 1996*; *Simkin et al., 2022*). The results of the study showed that Pn decreased with the intensification and duration of drought stress at the flowering and grain-filling stages of soybean. The decrease in Pn under drought stress at the flowering stage was dominated by stomatal factors. When under drought stress, it could reduce water evaporation by closing stomata, and the decrease in stomatal conductance directly reduced the rate of $CO_2$ entering the plant from the outside, thus leading to a decrease in intercellular $CO_2$ concentration. However, the decrease in Pn under drought stress changed from the stomatal factor to the non-stomatal factor at the grain-filling stage. Under severe water stress, due to the water loss in cells and chloroplasts, the interstitial ion concentration of chloroplasts increases, which leads to the inhibition of some enzyme activities involved in carbon fixation in chloroplasts. Thus, the intracellular $CO_2$ concentration increases and the $CO_2$ concentration difference between the external environment and leaves decreases, resulting in a decrease in the net photosynthetic rate. Therefore, non-stomatal factors become the major factors limiting plant photosynthesis under severe drought conditions.

The effects of drought stress on the activities of antioxidant enzymes, osmotic adjustment substances, and photosynthesis are interrelated. When plants are under drought stress, the balance between electron generation and utilization maintained by the photosystem I complex is disrupted, leading to an excessive generation of excitation energy in the chloroplasts. This excessive accumulation of excitation energy triggers a series of harmful reactions. It significantly promotes the generation and accumulation of ROS, such as superoxide anions, hydrogen peroxide, and hydroxyl radical. These highly reactive molecules disrupt the delicate intracellular redox balance, thereby inducing oxidative stress. Oxidative stress ultimately leads to membrane lipid peroxidation. The MDA produced by drought stress can be an important indicator of the degree of membrane lipid peroxidation, causing damage to the cell membrane, thus damaging plant tissues, accelerating plant senescence, and causing plant death in severe cases (*Osakabe et al., 2014*). Under drought stress, the MDA content in soybean increased rapidly, which is consistent with the results of this study. During this process, the unsaturated fatty acids in the cell membrane are attacked by reactive oxygen species, causing the degradation of membrane lipids. This not only impairs the integrity and fluidity of the cell membrane but also disrupts its normal functions. Meanwhile, DNA will also be damaged. Reactive oxygen species can interfere with the replication and transcription of DNA, potentially leading to abnormal gene expression. In addition, the photosynthetic system itself will also suffer from oxidative damage. Components of the photosynthetic apparatus, such as photosynthetic pigments, protein complexes, and electron carriers, will be damaged by reactive oxygen species, resulting in a decrease in photosynthetic efficiency, a reduction in carbon fixation, and ultimately, the inhibition of plant growth and yield (*Moldovan & Moldovan, 2004*).

Drought stress enabled plant protective enzyme systems, such as SOD and POD, to effectively remove reactive oxygen species and protect the plasma membrane, in which SOD activity played a leading role and was positively correlated with the antioxidant stress ability of plants (*Chaves, Maroco & Pereira, 2003*). However, the activity of antioxidant enzymes did not increase all the time, and if the intensity of drought stress exceeded a certain threshold, the activity of the enzyme decreased (*Møller, Jensen & Hansson, 2007*). The results of the previous study showed that the MDA content showed a continuously increasing trend under drought stress, and the increase in MDA content accelerated with the increase in drought stress intensity and duration. Results of previous studies showed that the MDA content increased exponentially with the increase in drought stress intensity and duration and decreased significantly after rewatering (*Del Longo et al., 1993*; *Scandalios, 1993*; *Zhang et al., 2015*), which was consistent with this study. The membrane protection system in plants can remove excess free radicals, among which SOD and POD are the main antioxidant enzymes. SOD can catalyze the transformation of superoxide anion ($O_2^-$) into hydrogen peroxide ($H_2O_2$), and POD can convert $H_2O_2$ into $H_2O$. They play a key role in the biological process of removing excessive reactive oxygen species in plants (*Zia et al., 2021*; *Asada, 2006*; *Miller et al., 2010*; *Kumar, Ayachit & Sahoo, 2020*). Drought resistance of plants is related to the ability of the protective enzyme system to remove reactive oxygen species. In the process of drought stress, with the increase in stress intensity, the activities of protective enzymes, SOD and POD, increased under mild drought stress. The increase in enzyme activity means that plants can more efficiently convert superoxide anion radicals into relatively stable hydrogen peroxide, thereby reducing the risk of oxidative damage caused by superoxide anion radicals to cells and maintaining the metabolic balance of reactive oxygen species within the cells. But the activities of both protective enzymes decreased under severe drought stress, in which SOD activity decreased faster and POD slower. From a mechanistic perspective, severe drought stress may severely disrupt the metabolic processes within plant cells, affect the expression of genes related to SOD and POD as well as the synthesis of corresponding proteins, or damage the structure of enzyme proteins, leading to a decrease in their activities. It indicated that SOD activity played an important role in resisting mild drought stress, and POD activity played a greater role in resisting severe drought stress (*Niu et al., 2018*). This study also showed that the SOD activity increased significantly under mild drought stress but decreased under severe drought stress. The increase in SOD activity under 7 and 14 days of mild drought stress was higher than that under 21 days, indicating that SOD activity can be inhibited with the increase in drought intensity. In terms of POD activity, in this study, only 7 days of mild drought at the grain-filling stage showed an increasing trend, while other treatments showed a decreasing trend. Previous studies have shown that POD activity increased under mild drought stress and decreased under severe drought stress (*Ahmadi, Emam & Pessarakli, 2010*). It was speculated that POD activity decreased due to active oxygen generation at the late stage of drought stress.

Plants can mitigate the damage caused by stress by metabolizing osmoregulatory substances. Common osmotic regulatory substances are divided into two categories: the first contains inorganic ions, such as $K^+$, $Na^+$, $Ca^{2+}$, and the other consists of organic

substances, such as proline, soluble sugar, and soluble protein (*Blum, 2017*; *Ahmed et al., 2017*; *Bai et al., 2019*). When plants are subjected to drought stress, the photosynthetic rate of leaves decreases, the ability of photosynthate transformation weakens, and the accumulation of carbohydrate substances occurs, resulting in the breakdown of osmotic regulation balance, thus inhibiting normal growth and development (*Ding et al., 2017*). The results of this study showed that the soluble protein content continued to increase with the increase in drought intensity and duration. Previous studies have shown that in the early stage of drought stress, the soluble protein accumulates at a faster rate, while in the late stage of drought stress, the soluble protein accumulates at a slower rate (*Ranney, Whitlow & Bassuk, 1990*), which is consistent with the results of this study that the increase in soluble protein content during 21 days of drought stress was less than that during 7 and 14 days of drought stress. After rewatering, osmoregulatory substances, such as proline and soluble protein, which are accumulated under drought stress, are the nitrogen sources that plants can use directly and serve as the material basis of the post-drought compensation effect (*Filippou et al., 2014*).

The post-drought rewatering showed that water supply restoration after a certain degree of drought stress could stimulate the physiological compensation effect of soybean plants, and the recovery of physiological characteristics depends on the occurrence time, degree, duration of stress, and drought-resistant ability of plants. The results of this study showed that the soluble protein and MDA contents and POD and SOD activities of soybean showed compensatory effects after rewatering under drought stress. Therefore, drought can increase the activities of antioxidant substances and the contents of osmotic regulatory substances in soybean leaves, and rewatering can alleviate the damage caused by drought stress. In addition, different intensities of drought stress show different degrees of compensation effects. The mechanism of the compensation effect involves osmotic adjustment, photosynthetic compensation, and ROS scavenging. Rewatering after drought enables plants to maintain a relatively high osmotic adjustment capacity for an extended period to compensate for the losses caused by drought (*He et al., 2024*). In terms of photosynthetic efficiency, rewatering can keep the stomatal aperture at a low level and alleviate photoinhibition (*Escalona, Flexas & Medrano, 2000*). After drought stress and subsequent rewatering, the activities of enzymes such as POD and SOD can remain at a relatively high level for a certain period to enhance the ability to scavenge ROS (*Mu et al., 2021*).

Soybean yield components are sensitive to soil water supply deficit (*Gao et al., 2018*). In this study, with the increase in drought stress intensity and duration, the number of pods and grains per plant decreased, the number of blighted grains per plant increased, and the yield of soybean decreased. The grain-filling stage is a crucial period for the formation and enrichment of soybean grains, during which photosynthetic products need to be transported to the grains for accumulation. Drought stress will affect photosynthesis, reducing the synthesis of photosynthetic products. At the same time, it will also interfere with the transportation and distribution of photosynthetic products, preventing the grains from receiving sufficient nutrient supply. As a result, the grain weight is reduced, and the yield decreases significantly (*Poudel et al., 2024*). During the flowering stage, drought

may affect the development of flowers and the pollination process. However, at this time, the growth centers of soybeans are mainly flowers and young pods, and the plants have relatively strong adaptability to drought (*Poudel et al., 2023*). Therefore, the impact on yield is relatively smaller compared with that during the grain-filling stage. Previous studies have shown that drought stress affects the proportion of material distribution in various organs and then affects reproductive growth and development, resulting in a decrease in yield (*Fernie et al., 2020*; *Li et al., 2018*; *Fawen, Manjing & Yaoze, 2022*; *Shemi et al., 2021*). In this study, rewatering after drought stress at different growth stages showed that the change in Pn was consistent with the yield because photosynthesis is the basis of crop dry matter accumulation and yield formation. In this study, Pn at the grain-filling stage was more sensitive to drought stress; thus, drought stress had a greater effect on soybean yield at this growth stage.

## CONCLUSIONS

This study clearly demonstrates that drought stress is a major determinant of reduced soybean yields. As the duration and intensity of the stress increased, the yield reduction became more significant, and the impact of drought on yield was greater during the grain-filling stage than during the flowering stage. Under mild drought stress conditions, the activity of SOD in soybeans increased, while severe drought reduced its activity. During the flowering stage, drought increased the activity of POD in soybeans, and the POD activity continued to rise after rewatering. However, during the grain-filling stage, the POD activity decreased after drought treatment. Drought stress induced an increase in the contents of MDA and soluble proteins in soybean leaves. After rewatering, the differences in the contents of MDA and soluble proteins between each treatment and the control gradually decreased. As the duration and intensity of the stress increased, the changes in the above physiological indices became more drastic. After rewatering, different levels of compensation were observed in each physiological index parameter. With the increase in the level and duration of drought stress, the compensatory effect also changed. In addition, the impact of drought stress on the Pn was mainly caused by stomatal conductance. However, during the grain-filling stage, as the duration of the stress increased, the influencing factors of Pn began to shift, and non-stomatal limitations started to dominate. Overall, these findings underscore the complex interplay between drought stress, physiological adjustments, and photosynthetic processes in soybeans, providing crucial insights for developing strategies to enhance soybean resilience to drought.

### Funding

This work was funded by The National Key Research and Development Program of China (2023YFD1500704). The funders had no role in study design, data collection and analysis, decision to publish, or preparation of the manuscript.

## Grant Disclosures

The following grant information was disclosed by the authors:

The National Key Research and Development Program of China: 2023YFD1500704.

## Competing Interests

The authors declare there are no competing interests.

## Author Contributions

- Cheng Wang performed the experiments, analyzed the data, prepared figures and/or tables, authored or reviewed drafts of the article, and approved the final draft.
- Anni Sun performed the experiments, analyzed the data, prepared figures and/or tables, and approved the final draft.
- Li jie Zhu performed the experiments, analyzed the data, prepared figures and/or tables, and approved the final draft.
- Min Liu analyzed the data, authored or reviewed drafts of the article, and approved the final draft.
- Qi Zhang analyzed the data, authored or reviewed drafts of the article, and approved the final draft.
- Liwei Wang conceived and designed the experiments, performed the experiments, analyzed the data, prepared figures and/or tables, authored or reviewed drafts of the article, and approved the final draft.
- Xining Gao conceived and designed the experiments, analyzed the data, authored or reviewed drafts of the article, and approved the final draft.

## Data Availability

The raw data is available in the Supplemental Files.

## Supplemental Information

Supplemental information for this article can be found online at http://dx.doi.org/10.7717/peerj.19658#supplemental-information.

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
