# Peer review of "Drought and rewatering effects on soybean photosynthesis, physiology and yield"

_PeerJ, doi:10.7717/peerj.19658_

## Round 0.1 · original submission · Major Revisions

The manuscript must be significantly improved. Revise per reviewers' suggestions.

Reviewer 1 ·

Basic reporting

The manuscript investigated the photosynthetic and physiological characteristics and yield of Glycine max during different growth stages under drought stress and rewatering. The results showed that the stomatal limit value increased and intercellular CO2 concentration decreased with the increase in drought stress intensity, and the decrease in net photosynthetic rate was dominated by stomatal factors at the flowering stage. The manuscript is well written and has a technical sound. However, some issues need to be addressed before the paper can be accepted for publication as follows:

1) The title is precise but slightly long. Simplifying it to focus on key elements

2) The abstract mentions the application of drought stress at the flowering and grain-filling stages. However, it does not specify the intensity or duration of the drought stress applied. Providing some context or quantitative values related to stress levels would enhance the reader's understanding of the experimental design.

3) In the introduction, the authors should provide details regarding the interactions between drought stresses and plants, and the mechanisms that may be adapted to alleviate the harmful effects of drought on plants.

4) Please add one sentence to each paragraph in the results to make it easy for the readers to follow.

5) The discussion provides valuable interpretations but tends to repeat the results without expanding on broader implications or comparisons with other studies. Adding more context and literature comparisons would strengthen the section.

6) The conclusions need to be restructured to show the novelty and strengths of the study.

Experimental design

-

Validity of the findings

-

Reviewer 2 ·

Basic reporting

No comment.

Experimental design

- A field experiment implemented only for one year is, to the best of my knowledge, not enough to draw conclusions from.
- Line 108: what is “random block groups”? do you mean a completely randomized design? Or a randomized complete block design? Or what exactly?

Validity of the findings

- The main concern is the importance of the experiment: it’s been widely reported that reproductive stages are more drought-susceptible to drought than vegetative stages, especially the flowering and pod filling stages. The claim that “Most studies adopted pot experiments that achieved drought stress in a relatively short time” (lines 68-69) is not accurate (please do some basic search in Scopus or Web of Science databases).

Additional comments

- Latin names should be italic.

Reviewer 3 ·

Basic reporting

It is understandable; however, needs more refinement.

Experimental design

The experimental design should be more elaborate to replicate the experiment.

Validity of the findings

This is not a novel study.

Annotated reviews are not available for download in order to protect the identity of reviewers who chose to remain anonymous.

·

Basic reporting

The manuscript titled "Effects of drought stress and rewatering on photosynthetic and physiological characteristics of soybean” investigates the photosynthetic and physiological characteristics, as well as the yield of Glycine max (soybean) during different growth stages under drought stress and rewatering. The authors conclude that drought has the greatest impact on soybean yield during the grain-filling stage, highlighting the associated physiological changes.
While I appreciate the authors' efforts and recognize the importance of this timely research, I have several specific concerns regarding the manuscript, as outlined below.

Abstract:
Line 18: Authors have mentioned “Growth and Development”; however, it is important to clearly mention in the subsequent text, probably in the methodology, about the growth events and development events that authors focused on.
Authors need to be carefully select appropriate words to convey the message. I have found several instances where authors have used some words that do not convey the required meaning. Following are some instances along with potential suggestions for corrections.
Line 40: the word “affect” needs to be corrected as “creates”
Line 45: the word “important” needs to be corrected as “main/major”
Line 68: the word “conditions” needs to be corrected as “studies”

Introduction:
The whole introduction goes as a single paragraph; however, to improve the clarity and readability it can be separated into several paragraphs, probably as follows.
Line 50-55 : Rewatering can be a separate paragraph.
Line 55: A new paragraph can be started with “Severe drought stress.”

In addition to the above, authors need to attend to the following in order to improve the ms.
Line 63: “Previous studies” please provide some references to support this claim
Line 72: “Previous studies” please provide some references to support this claim
Line 74: “ Different times” can be changed to “the time at which drought occurs”
Line 90: Bulk density – given units seems incorrect and it should be g cm-3,
“Field water capacity was 30%” - what is mean by this? Is this 30% of the field capacity?
Line 99: “Soya bean soil moisture” should be corrected as “soil moisture.”
Line 108: “Random block groups” what is meant by this in not clear? Please clarify
Line 109: “ Row spacing and plant spacing’ I guess what authors mentioned could be inter-row spacing and intrarow spacing.
Line 114: “Plants were continuously taken” The idea of this is not clear. Please clarify in the text.
Line 118: “Measurements were performed the day after drought stress treatments” what is meant by this is not clear. I suppose this should be “measurements were made the day after the duration of drought stress treatments were completed.”
Line 128: State the temperature at which the photosynthesis measurements were made.
Line 137: Apparent Quantum Efficiency should be denoted as AQE and not as ACE
Line 145: “low temperature treatments “There was no indication regarding the use of low temeparture in this experiment. Can you please clarify.

Results:
Line 297: “non stomatal conductance” should be corrected as non stomatal limitations
Line 336-337: The sentence, “Therefore, non-stomatal factors become the important factors limiting plant photosynthesis” appears incomplete. To improve clarity and precision, consider adding “under severe drought conditions” at the end, revising it to:
“Therefore, non-stomatal factors become the major factors limiting plant photosynthesis under severe drought conditions”

Discussion: This section has been written well.

Conclusion:
Line 420: “Morphological index” – authors haven’t either defined or mentioned this in the text, and all of sudden this has appeared in the discussion. Please clarify
Line 421-427: The conclusion section is overly generalized. Since plant responses to stress vary significantly across species, the conclusions should specifically address soybean (Glycine max) rather than making broad statements about "plants" in general.

Experimental design

Experimental Design
The authors mentioned that plots were arranged in random groups, which is not clear as no such statistical design is available. Instead authors need to mention what the correct design used for laying the treatment.

Data analysis:
The authors have used one-way ANOVA for data analysis; however, according to the results presented, I found two main factors ie. Watering (well water, 65% FC and 50% FC) and duration of drought exposure ( 7, 14 & 21 days). It seems like 2-way ANOVA is most appropriate since you can find interactions among the two factors and easily explain how one treatment varies in response to different levels of the other factor.
Further, I would suggest that authors explain the statistical analysis with more details to improve the understanding of the reader.

Validity of the findings

Methodology: The photosynthesis measurement protocol (9:00-12:00) lacks crucial information about leaf temperature monitoring. Given the 3-hour measurement window and potential temperature fluctuations, this omission raises serious concerns about the validity of the data presented and the validity of findings, particularly for a study examining stress responses.

The reference to "low temperature treatments" appears inconsistent, as this factor is not addressed elsewhere in the manuscript. Either these treatments should be properly integrated into the methodology and results or removed to avoid confusion.

Data Analysis: While the authors report using one-way ANOVA, this approach fails to examine important interaction effects between experimental factors. As noted in my specific comments to the authors, these interactions could be crucial for the proper interpretation of the results.

Dr. L.K. Weerasinghe

Additional comments

No additional comments to be made.

---

## Round 0.2 · Minor Revisions

In the revised version, some comments are not addressed. Revise the manuscript carefully. If you are not agree with any of the comments, write a justification.

Reviewer 1 ·

Basic reporting

Thanks a lot

The authors addressed all of my comments

Please accept the paper as it

Experimental design

Perfect

Validity of the findings

Wonderful

Additional comments

Thanks a lot

The authors addressed all of my comments

Please accept the paper as it

Reviewer 3 ·

Basic reporting

Need slight English refinement.

Experimental design

In the revised version, they included most of the necessary details.

Validity of the findings

Please include your field pictures as a main figure.

Additional comments

The authors made necessary corrections. After careful revision of above minor comments, it can be accepted.

·

Basic reporting

Line 18: Authors have mentioned “Growth and Development”; however, it is important to clearly mention in the subsequent text, probably in the methodology, about the growth events and development events that authors focused on.
According to your suggestion, we have revised the content of line 18 in the original manuscript, and the revised content is on lines 16-20 of the revised manuscript. In the part of research methods, we have also expounded that this study focuses on two important growth periods of soybeans, namely the flowering period and the grain-filling period.
Reviewer reponse: Abstract still indicates that “growth and development periods” in revised 16-20

Line 108: “Random block groups” what is meant by this in not clear? Please clarify
We deeply apologize. There was an error in the original expression of the part you mentioned. We have already revised the content of line 108 in the original text, and the new description is on line 168-171 of the revised version.
Reviewer reponse: Changes made by the authors can’t be found in the revised text

Line 128: State the temperature at which the photosynthesis measurements were made.
Thank you very much for your suggestion. In this study, the temperature was basically the same when measuring the photosynthetic indexes for different treatments. We focused on the impacts of drought treatments on the physiological and photosynthetic indexes of soybeans. We consider that temperature is not the main factor causing the differences among the treatments.
Reviewer reponse: If authors report photosynthetic values, they must also include the measurement temperature, as photosynthesis is temperature-dependent. Without this critical parameter, the interpretation of photosynthetic data remains incomplete.

Experimental design

The authors have used one-way ANOVA for data analysis; however, according to the results presented, I found two main factors ie. Watering (well water, 65% FC and 50% FC) and duration of drought exposure ( 7, 14 & 21 days). It seems like 2-way ANOVA is most appropriate since you can find interactions among the two factors and easily explain how one treatment varies in response to different levels of the other factor.
Thank you for your suggestion. In this study, the main purpose of using the two factors in the drought treatment (i.e., watering conditions and drought duration) was to achieve different degrees of drought. Therefore, we analyzed them as a single factor influencing drought.
Reviewer reponse: As noted earlier, the authors cannot establish an interaction between the two factors. If the interaction is significant, discussion of main effects becomes unnecessary.

Further, I would suggest that authors explain the statistical analysis with more details to improve the understanding of the reader.
Thank you for your suggestion. In this study, we only used Excel 2016 and SPSS25 software to conduct a relatively uncomplicated one-way analysis of variance (ANOVA) to determine whether there were significant differences in various indicators among different drought treatments and their changing trends. This part has been described in relatively detail in the research methodology section. More in-depth statistical analyses are yet to be carried out in the future, such as principal component analysis of different influencing factors, etc.
Reviewer reponse: The authors should remove Line 202 ('Excel 2016 was used to sort the test data and draw charts') from the revised manuscript, as such procedural details about data organization and graph preparation are unnecessary.

Validity of the findings

Line 421-427: The conclusion section is overly generalized. Since plant responses to stress vary significantly across species, the conclusions should specifically address soybean (Glycine max) rather than making broad statements about "plants" in general.
Thank you for your valuable comment regarding the restructuring of the conclusions. We have restructured and summarized the conclusions of this study. The revised contents are located in lines 507-522 of the revised manuscript.
Reviewer reponse: Stating the conclusion from the study with a clear take-home message to the reader is important here. So, I would suggest that authors can do a little bit of revision here again.

Additional comments

The authors have addressed the majority of my previous comments. However, I have provided additional feedback under “reviewer response” on the remaining issues that require attention. Once these revisions are made, I would consider the manuscript suitable for publication.

Dr. L.K. Weerasinghe

---

## Round 0.3 · Minor Revisions

Mention the temperature used for measurement of photosynthesis.

·

Basic reporting

In response to my query, the authors state:

“In this study, the portable gas exchange measurement system (Li6400XT) we used is equipped with a temperature control device, which can regulate the temperature in the leaf chamber and maintain stability during measurements. Therefore, temperature fluctuations during the experiment were minimal.”

While I appreciate this clarification, I do not understand why the authors are reluctant to specify the actual temperature at which the photosynthesis measurements were made. This is a critical parameter for interpreting photosynthetic data, and omitting it significantly diminishes the usefulness of the reported values. The authors could simply state:
“Photosynthesis measurements were made at a block temperature of XX °C.”

Once the authors provide this essential detail, I will consider the manuscript suitable for publication.

Experimental design

OK

Validity of the findings

OK

Additional comments

No

---

## Round 0.4 · accepted · Accept

The authors have addressed all the comments. The manuscript can be accepted for publication.

·

Basic reporting

Ok

Experimental design

OK

Validity of the findings

OK

Additional comments

Now the ms can be accepted for publication.